# The Mediating Role of Psychological Resilience in the Relationship between Emotional Reactivity, Intolerance of Uncertainty and Psychological Maladjustment in Children Receiving Orthodontic Treatment

**DOI:** 10.3390/healthcare10081505

**Published:** 2022-08-10

**Authors:** Nurhan B. Durna, Doğan Durna, İsmail Seçer

**Affiliations:** 1Faculty of Densitry, Ataturk University, Erzurum 25240, Turkey; 2Department of Psychology, Ataturk University, Erzurum 25240, Turkey

**Keywords:** orthodontic treatment, emotional reactivity, intolerance of uncertainty, psychological adjustment, mediation relationship

## Abstract

The aim of this research was to examine the various psychological characteristics that affect psychological maladjustment in children undergoing orthodontic treatment. In this context, the predictive and mediating relationships between emotional reactivity, intolerance of uncertainty, psychological resilience and psychological maladjustment were considered. The study was conducted cross-sectionally with 543 children and adolescents aged 10–18 years, who were being treated at a state university orthodontic clinic in Turkey. Standardized measurement tools (The Emotional Reactivity Scale, Intolerance of Uncertainty Scale, Brief Resilience Scale and Depression Anxiety Stress Scale) and online data collection processes were used in the data collection process. The findings show that emotional reactivity and intolerance of uncertainty pose a risk for psychological maladjustment in children and adolescents receiving orthodontic treatment, but psychological resilience has a protective function against this risk (*p* < 0.001). It is suggested that these findings may contribute to the expansion of pediatric dentists’ perspectives on the secondary outcomes of orthodontic treatment practices.

## 1. Introduction

Orthodontic treatment is a relatively long and laborious treatment process that is used in the treatment of common dental ailments today, and the individuals who apply for this treatment are mostly children and adolescents. The treatment process is usually spread over a long period of 1–2 years. For this reason, it is possible that some psychological consequences may occur in orthodontic patients who are faced with a difficult treatment process [1,2,3]. In this context, it is thought that uncertainty regarding the treatment process, the fear that occurs due to orthodontic treatment and reflections on the apparatus used during the treatment may trigger the aforementioned secondary psychological results.

Individuals’ reactions to difficult living conditions are generally defined as shock, panic, acute stress, post-traumatic stress disorder, anxiety disorder and depression, etc. [4] and these symptoms are impacted by the individual’s psychological adjustment skills. Psychological adjustment can be defined as the ability of the individual to cope with daily life difficulties, to control intense anxiety, depressive symptoms and stress factors, and it is thought that difficult living conditions have an effect that challenges the psychological adjustment skills of the individual. Accordingly, long and troublesome orthodontic treatments can put pressure on these symptoms and can challenge the individual’s adjustment skills. It is also possible that the traumatic fear experienced by children and adolescents due to COVID-19 during the last year [5,6] might intensify the risk factors that develop due to orthodontic treatment and put these individuals in a disadvantaged position.

It is possible for orthodontic treatment to cause fear, depending on the age of the child. It has been suggested by Ornell et al. [7] and Shigemura et al. [8] that fear, which is a defense mechanism that the individual shows in the face of dangerous situations, when disproportionate to the conditions of the individual, may pave the way for various psychological disorders such as anxiety, depression, stress and OCD (Obsessive Compulsive Disorders), etc. [9,10]. Therefore, the fear that may arise during the orthodontic treatment process can be thought to present various risks in terms of psychological adjustment. Factors naturally resulting from the COVID-19 pandemic, namely negativities experiences in treatment planning due to quarantine and social distancing practices, are also thought to increase such risks in children and adolescents, and may intensify their disadvantaged position [5]. Research results [6,11] appear to support this view, showing that the fear of catching COVID-19 causes intense emotional and behavioral consequences such as boredom, loneliness, anxiety, sleep problems and anger. Thus, the psychological symptoms that individuals develop in relation to orthodontic treatment might be worsened due to the effects of the risk factor created by the COVID-19 pandemic.

It is also thought that some psychological qualities of children may increase the likelihood or severity of secondary results due to orthodontic treatment. Among these qualities, the intensity of the emotions experienced by the individual in various situations, and their emotional reactivity, which defines the reactions that are elicited as a result of this intensity, can be identified [5,12]. High emotional reactivity is thought to pave the way for the development of psychological symptoms related to the treatment process in children. The literature data show that high emotional reactivity is associated with major depression [13], anxiety disorders [5,14] and OCD symptoms [15]. No studies addressing the role of emotional reactivity in orthodontic and dental treatments have been found in the literature. Therefore, it is important to assess the role of this variable when attempting to understand the possible risk factors in orthodontic treatment planning.

Another risk factor for the secondary results of orthodontic treatments is *intolerance of uncertainty*. İntolerance of uncertainty is defined as a tendency to react emotionally, cognitively and behaviorally to uncertain situations and events [2]. It has been reported that people with high intolerance of uncertainty tend to see uncertain situations as annoying and stressful, to avoid this uncertainty and to experience difficulties in their functioning in situations involving uncertainty [3,16,17]. It has also been argued that individuals’ perceptions and interpretations of uncertain situations contain a negative bias, meaning these people are more prone to interpret uncertain situations as threatening [16]. Since orthodontic treatment is a long-term process and involves uncertainty regarding the duration and success of the treatment, intolerance of uncertainty is an important risk factor in terms of the psychological symptoms that may accompany it.

In addition to these negative characteristics of individuals receiving orthodontic treatment, there are also individual qualities, such as psychological resilience, which have a protective function. *Psychological resilience* is defined as the ability of an individual to recover quickly in the face of difficult living conditions, and to return to their former state after being injured [11,18,19,20]. Similarly, this quality has also been defined as the ability of an individual to succeed in the face of uncertain and challenging processes [18] and to quickly regain the ability to fulfill the duties and behaviors expected of them [21]. From this point of view, it can be said that psychological resilience is an important protective feature to consider when attempting to reduce the risk of psychological symptoms caused by emotional reactivity and intolerance of uncertainty in orthodontic treatment.

### The Current Study

This study sets out to examine various psychological variables that predict psychological maladjustment and mediate these predictive relationships in children receiving orthodontic treatment. In this context, the mediating role of psychological resilience in the predictive relationship between emotional reactivity, intolerance of uncertainty and psychological maladjustment was examined. This research aimed to expand on current perspectives and add to the limited number of studies in the literature that examine the psychological consequences of orthodontic treatments. It did so by analyzing the personal risk factors and protective factors that impact the negative psychological effects on children receiving orthodontic treatments that require a long-term treatment plan.

Within this scope, the research questions to be answered were as follows:Are emotional reactivity and intolerance of uncertainty a significant predictor of psychological maladjustment in children and adolescents receiving orthodontic treatment?Is there a mediating role of psychological resilience between emotional reactivity, intolerance of uncertainty and psychological maladjustment in children and adolescents receiving orthodontic treatment?

## 2. Materials and Methods

### 2.1. Participants

The research process was carried out with children and adolescents who were under treatment at Atatürk University Faculty of Dentistry Orthodontic clinic. Study participants were reached through systematic and appropriate sampling. The inclusion criteria of the research sample were as follows: being between the ages of 10–18, receiving continuing orthodontic treatment, not having received orthodontic treatment before, not having congenital anomalies, not having systemic disease and not having a dentofosial deformity caused by trauma. When calculating the sample size during the research process, the number of patients treated in the relevant clinic within one year was taken as a reference. In this context, the sample calculation was made based on a 95% confidence interval over the records in the database. When possible lost data etc. were taken into account, the estimated figure was exceeded. The data collection process was completed in 2021. Following this, the research process (m = 15.30, Sd = 2.14) was carried out with a total of 543 children and adolescents. Of the total participants, 56.42% received fixed treatments, 18.9% received orthognathic surgery, 13% received fixed treatments with an extraoral device and 11.7% received a mobile apparatus treatment.

### 2.2. Measures

In this research process, data collection tools with proven reliability and validity were used in line with the purposes of the research. Introductory information about these measurement tools is presented below.

#### 2.2.1. The Emotional Reactivity Scale

The Emotional Reactivity Scale was developed by Nock, et al. [12] to measure the emotional intensity experienced in the face of situations that arise in interpersonal relationships and the reactivity expressed in these intense emotional situations; it was subsequently adapted to the Turkish context by Seçer et al. [22]. The scale is a self-report and four-point Likert scale. It includes a total of 15 items and 3 sub-dimensions, and its original 3-factor structure was preserved during the Turkish-culture adaptation process. It was determined that the reliability values for the sub-dimensions varied between 0.81 and 0.94. During this study, the construct validity of the scale was revised as follows: χ^2^/sd = 1.96, REMSEA (The Root Mean Square Error of Approximation): 0.062; RMR (Root Mean Square Residual): 0.063; SRMR (Standardized Root Mean Square Residual): 0.067; CFI (Comparative Fit Index): 0.98) and it was determined that the fit indices were at a good level. The current internal consistency value of the scale was calculated as 0.87. High scores from the scale indicate that emotional reactivity is at a high and risky level.

#### 2.2.2. Intolerance of Uncertainty Scale

The Intolerance of Uncertainty Scale is a four-point Likert-type measurement tool developed to measure the susceptibility of individuals to exhibit negative emotional, cognitive and behavioral reactions to uncertain events and situations [23]. The scale consists of 27 items and four sub-dimensions. It was determined that the scale preserved the structure of the original form following its adaptation to the Turkish cultural context, with a structure explaining 48% of the variance and an internal consistency coefficient of 0.87 [24]. During this study, the construct validity of the scale was revised (χ^2^/sd = 1.96; REMSEA: 0.062; RMR: 0.063; SRMR: 0.067; CFI: 0.98) and it was determined that the fit indices were at a good level. The current internal consistency value of the scale was calculated as 0.82. High scores from the scale indicate that individuals show cognitive and affective intolerance in situations involving uncertainty.

#### 2.2.3. Brief Resilience Scale

The Brief Resilience Scale is a four-point Likert (never, rarely, often, always) measurement tool developed by Smith et al. [25] and adapted to Turkish culture by Doğan [26]. The scale consists of 6 items in total, and high scores indicate high psychological resilience. The scores that can be obtained from the scale range from 6 to 24. Scale items include, for example, “I tend to bounce back quickly after hard times” and “I usually come through difficult times with little trouble”). In this study, the construct validity of the scale was reviewed, and it was determined that the model fit indices (χ^2^/df = 1.96; REMSEA: 0.062; RMR: 0.063; SRMR: 0.067; CFI: 0.98) were at a good level. The Cronbach alpha internal consistency value was calculated as 0.87.

#### 2.2.4. Depression Anxiety Stress Scale

The Depression Anxiety Stress Scale is a four-point (never, rarely, often, always) Likert-type measurement tool consisting of 42 items, and was developed by Lovibond and Lovibond [27] to measure depression, anxiety, and stress symptoms. It was later revised by Brown et al. to include 21 items [28]. The scale was adapted to Turkish by Yılmaz et al. [29]. The data regarding the construct validity of the scale (χ^2^/df = 2.84; REMSEA: 0.051; RMR: 0.036; CFI: 0.98) showed that the three-factor structure consisting of 21 items had a good level of fit. Questions include the following: “I felt scared without a valid reason” and “I was worried about situations where I would panic and make myself stupid”). The scores obtained from the scale range from 21 to 84, and high scores indicate high levels of depression, anxiety and stress symptoms.

### 2.3. Procedure and Data Analyses

The research process began with the approval of the Ethics Committee of the Faculty of Dentistry, Atatürk University for research compliance. The data collection process was carried out using online tools due to COVID-19 social distancing restrictions. In this context, children who were registered in the hospital database and who were receiving continuing orthodontic treatment were identified. In the second stage, the junior doctors who followed the treatment processes of these children were contacted to reach the children and their parents, and their consent was obtained for voluntary participation. The parents of the children who volunteered to participate were asked to help their children complete the measurement tools, which were sent as a link via e-mail and WhatsApp-like applications. An online data collection link prepared through Google Documents was used for this purpose (available from https://forms.gle/xJRQ3krHVo5cfDzu7 accessed on 1 March 2021). Additional explanations regarding study participation and data privacy were also included in this link, along with the instruction that participants were free to withdraw from filling in the questionnaire at any time. The data collection process was completed within 20 days. Data collection and compilation procedures were carried out by three different specialists from dentistry, orthodontics and psychology. It was determined that the data of 21 participants in the data set did not meet the normality-homogeneity criteria, and consequently these responses were excluded from the analysis.

During the analysis process, structural equation analyses were carried out with the LISREL 9.2 software. The confirmatory measurement model test was conducted during the first stage of the analysis process and it was determined that the designed model had a good fit (χ^2^/sd = 1.60; REMSEA: 0.071; RMR: 0.073; SRMR: 0.073; NFI: 0.95; CFI: 0.97; GFI: 0.92). Verification of the measurement model showed that all the implicit variables in the model had a good level of fit with the indicator variables they represent and the other implicit variables [30]. After the measurement model, three different structural models were tested for the purposes of the research. CFI, NFI, GFI, RMR, SRMR, RMSEA and χ^2^ values were examined as fit indices in the structural equation model. In the evaluation of the model fit indices, different criteria were taken into account as suggested. Specifically, the model fit indices in the structural equation model should be 0.90 for an acceptable fit and ≥0.95 for a perfect fit for RFI, TLI, CFI, NFI, NNFI and IFI; ≥0.85 for an acceptable fit and ≥0.90 for a perfect fit for GFI and AGFI; and ≤0.08 for an acceptable fit and ≤0.50 for a perfect fit for RMR, REMSEA and SRMR [31].

## 3. Results

Three different structural models were tested in line with the research questions. Each model and its findings are presented below. In this context, the research hypothesis was set as follows: “Emotional reactivity and intolerance of uncertainty predict psychological adjustment skills in children receiving orthodontic treatment”. This hypothesis was tested as Model 1. In this model, high emotional reactivity and high intolerance of uncertainty were expected to positively predict psychological adjustment skills in children receiving orthodontic treatment. Findings related to Model 1 are presented in Figure 1. 

Considering the fit index values χ^2^ (44.26/34) = 1.30; CFI = 0.97; TLI = 0.96; NFI = 0.94; GFI = 0.93) for the model tested in Figure 1, it can be said that all of the implicit variables in Model 1 have a significant relationship with the observed variables they represent (*p* < 0.001). In this sense, it is seen that emotional reactivity (β = 0.41, *p* < 0.01, 17%) and intolerance of uncertainty (β = 0.47, *p* < 0.01, 22%) are positive and significant predictors of psychological adjustment skills in children receiving orthodontic treatment. Although the relationship patterns determined between the variables are significant and high, it is also recommended to include the variables that are likely to mediate these relationships in structural equation models and to test their effect. Therefore, the possible mediation of the relationship patterns determined in Model 1 was analyzed by including the psychological resilience variable in the model. In this process, defined as Model 2, the direct and indirect effects of emotional reactivity and intolerance of uncertainty on psychological adaptation skills were examined. In this sense, the research hypothesis, constructed as Model 2, was expressed as follows: “How did the direct prediction effect of emotional reactivity and intolerance of uncertainty on psychological adaptation skills in children receiving orthodontic treatment change after the inclusion of resilience in the model?”. The findings obtained for this model are presented in Figure 2.

When Model 2, in which mediation relations are tested, is examined, it can be seen that the mediation of psychological resilience is significant and the fit indices are sufficient. The general rule in mediation relations is that when the “mediator variable” is included in the model, a significant change should occur in the direct prediction coefficients obtained in Model 1. When Model 2 is examined, there is no significant change in the predictive coefficients of emotional reactivity and intolerance of uncertainty obtained for Model 1 with regard to psychological maladjustment. However, one striking point in Model 2 is the finding that the relationship between psychological resilience and psychological adjustment was not significant, though emotional reactivity (β = 0.47, *p* < 0.01, 22%) and intolerance of uncertainty (β = 0.47, *p* < 0.01, 22%) negatively predicted psychological resilience. This phenomenon is the result of a Type II error, which stems from the direct prediction paths in the model. Consequently, a new model was devised to fully test the mediator relations. In this model, defined as Model 3, the answer was sought for the research question expressed as “*Does psychological resilience fully mediate the relationship between emotional reactivity and intolerance of uncertainty and psychological adjustment?*” The findings regarding this model are presented in Figure 3. This model aimed to prevent a Type II error and analyze the real relationship patterns between the variables by removing the direct paths from emotional reactivity and intolerance of uncertainty from the model.

When Figure 3 is examined, it can be seen that the model that tests the full mediation of psychological resilience in children receiving orthodontic treatment is well adapted and significantly differentiated from Model 2. In addition, there is a significant improvement in the prediction coefficients and fit indices between the variables compared to Model 2 (χ^2^/sd (299.32/205) = 1.46; CFI = 0.98; TLI = 0.97; SRMR = 0.048; RMSEA = 0.046). When the findings obtained regarding the mediation model were examined, it was found that emotional reactivity (β = −0.67, *p* < 0.01, 45%) and intolerance of uncertainty (β = −0.24, *p* < 0.01, 6%) negatively predicted psychological resilience and it can also be observed that they predict psychological adjustment through resilience (β = −0.32, *p* < 0.01, 10%). The findings obtained in Model 3 contain significant differences compared to Model 2. The first striking difference is that there is a significant increase in the predictive coefficients of emotional reactivity and intolerance of uncertainty on psychological resilience compared to Model 2. The second important difference is that, although the predictive effect of psychological resilience on psychological adjustment is insignificant in Model 2, a serious change occurred in this predictive coefficient in Model 3 (β= −0.78, *p* < 0.01, 60%). In this context, it can be said that psychological resilience has a fully mediating function in the relationship between emotional reactivity and intolerance of uncertainty and psychological adjustment in line with the findings in Model 3.

## 4. Conclusions and Discussion

In line with the findings obtained from the study, it can be determined that children and adolescents receiving orthodontic treatment have a high probability of developing psychological maladjustment (depression, anxiety and stress), that emotional reactivity and intolerance of uncertainty are risk factors for these symptoms and that psychological resilience stands out as an important variable that protects children and adolescents against this risk.

The research results show that emotional reactivity is a predictor of psychological maladjustment in children and adolescents receiving orthodontic treatment, and that high emotional reactivity creates a significant risk for psychological maladjustment. This finding, which is parallel with the related literature, is thought to be significant [5,12,13], since the long-term nature of orthodontic treatments, the potentially troublesome treatment process and the appearance of the apparatus used can possibly trigger psychological symptoms. Consequently, high emotional reactivity may increase the psychological symptoms in these children and cause the treatment to be negatively affected. It is even possible that emotional reactivity may have negative consequences to the point of interrupting and disrupting the treatment.

Another important finding of this research is the result showing that intolerance of uncertainty is a predictor of psychological maladjustment in children and adolescents receiving orthodontic treatment. The development of psychological symptoms appears to be more likely in children with a high level of intolerance of uncertainty [2,3,5]. It is possible that children with a low tolerance of uncertainty will develop more negative reactions in emotional, cognitive and behavioral terms, which may disrupt the process, affecting the duration and success of orthodontic treatment. 

The most striking finding of the study concerns the protective role of psychological resilience. The findings indicate that children who receive orthodontic treatment and have high psychological resilience are less likely to develop psychological symptoms. This finding, which coincides with the related literature [1,32,33,34], indicates that psychological resilience may play an important role in reducing the risk of emotional reactivity and intolerance of uncertainty in orthodontic treatment and in preventing children from developing psychological symptoms. This finding suggests that there is a need to consider these psychological processes at every stage of the orthodontic treatment procedure and to make a general evaluation in terms of psychological protective and risk factors, since such evaluations would provide important contributions both in preventing the development of treatment-related symptoms and in ensuring the continuity of the treatment.

Based on these results, it can be said that children with a low tolerance of uncertainty will develop more negative reactions in emotional, cognitive and behavioral terms, which may affect the duration and success of orthodontic treatment. Furthermore, it would be beneficial to consider intolerance of uncertainty and emotional reactivity as prominent risk factors in orthodontic treatments, which may affect the course of the treatment. 

## 5. Limitations and Future Research

The findings of this research should be evaluated in the context of its limitations. The research was conducted only in a relational and cross-sectional context due to the negative circumstances created by the COVID-19 pandemic. In addition, and for the same reason, data collection was completed online, and a convenience sample method was implemented. In the future, qualitative and mixed-based studies that focus on similar samples, including subjective evaluations of children and adolescents receiving orthodontic treatment, could make significant contributions to the literature. In addition, studies with an intercultural focus would strengthen the literature in this area. The effect of these factors on the research results should be considered.

### Why This Paper Is Important to Paediatric Dentists

It is believed that the results of the research will help expand current perspectives on orthodontic treatments carried out with children and young people in the national and international field and contribute to pediatric dentists’ perspectives on the secondary outcomes of such treatments. In addition, the study findings as to the protective and risk factors regarding psychological maladjustment in children receiving orthodontic treatment can contribute to the development of action plans for designing psychological intervention and therapy approaches.

## Figures and Tables

**Figure 1 healthcare-10-01505-f001:**
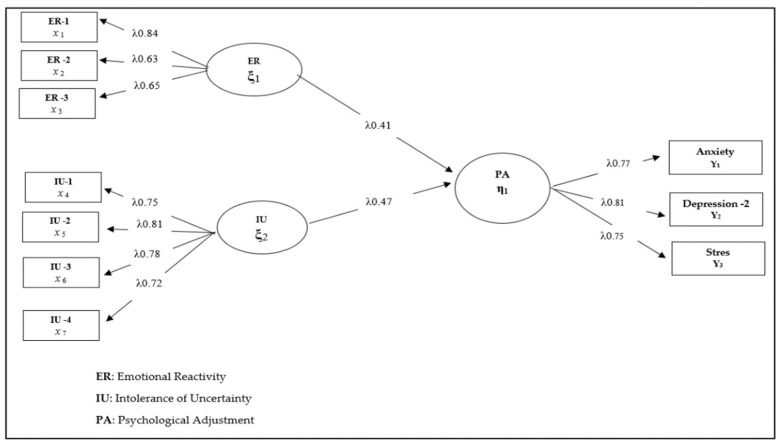
Standardized SEM results for Model 1. (Appendix A).

**Figure 2 healthcare-10-01505-f002:**
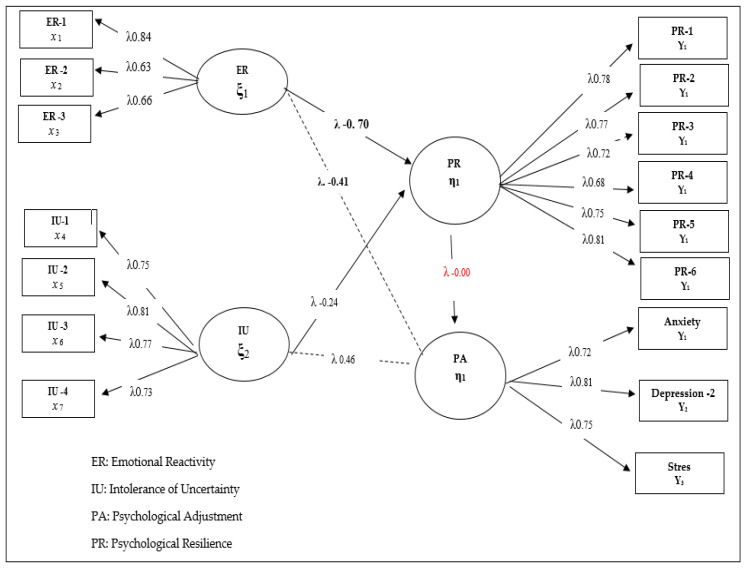
Standardized SEM results for Model 2. (Appendix A).

**Figure 3 healthcare-10-01505-f003:**
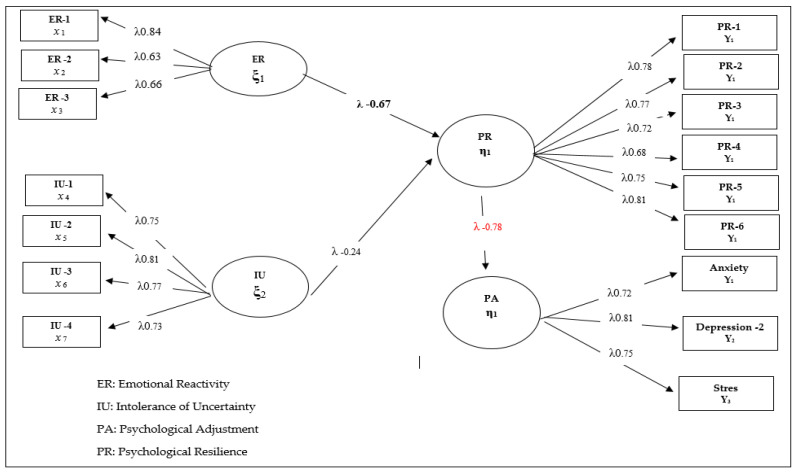
Standardized SEM results for Model 3. (Appendix A).

## Data Availability

Research data can be shared upon request.

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
