# Peer review of "The Mediating Role of Psychological Resilience in the Relationship between Emotional Reactivity, Intolerance of Uncertainty and Psychological Maladjustment in Children Receiving Orthodontic Treatment"

_healthcare, 2022, doi:10.3390/healthcare10081505_

Round 1
Reviewer 1 Report
This study aimed to examine various psychological variables that predict psychological maladjustment and mediate these predictive relationships in children receiving orthodontic treatment. It was examined the mediating role of psychological resilience in the predictive relationship between emotional reactivity, intolerance of uncertainty and psychological maladjustment. The problem is that since orthodontic treatments require a long-term process and the uncertainty about the duration and success of the treatment, it is an important risk factor in terms of psychological symptoms.
The research questions:1. Are emotional reactivity and intolerance of uncertainty a significant predictor of psychological maladjustment in children and adolescents receiving orthodontic treatment? 2. Is there a mediating role of psychological resilience between emotional reactivity, intolerance of uncertainty and psychological maladjustment in children and adolescents receiving orthodontic treatment?
The design of the study and methodology were chosen adequately to address the objectives. Four scales were used (The Emotional Reactivity Scale; Intolerance of Uncertainty Scale; Brief Resilience Scale and Depression Anxiety Stress Scale).
The data collection process was carried out using online tools due to COVID-19 social distance restrictions. In this context, children who were registered in the hospital database and whose orthodontic treatment continues were determined. In the second stage, the junior doctors who followed the treatment processes of these children were contacted to reach the children and their parents, and their consent was obtained for voluntary participation. Participants of the study consisted of 543 children aged between 10 and 18 (m = 15.30, Sd = 2.14) who were accessed through systematic and convenient sampling methods among the patients who were still being treated in the Orthodontics clinic of Faculty of Dentistry, Atatürk University.
The results clearly presented. Three different structural models were tested in line with the research questions. Research results show that emotional reactivity is a predictor of psychological maladjustment in children and adolescents receiving orthodontic treatment, and high emotional reactivity creates a significant risk for psychological maladjustment.
Conclusion. In line with the findings obtained from the study, it was determined that children and adolescents receiving orthodontic treatment have a high probability of developing psychological maladjustment (depression, anxiety, stress), emotional reactivity and intolerance of uncertainty are risk factors, and psychological resilience stands out as an important variable that protects children and adolescents against this risk.
The weaker part of the study was that research was conducted only in a relational and cross-sectional context due to the negativities created by the epidemic the heterogeneous distribution of one of the samples according to the career and work of the participants, which may undermine the conclusions regarding external validation. A strong part of the study was good methodology and the design of the study.
The manuscript is suitable for printing. It is recommended to print without corrections.
Author Response
Dear Rewiewer, Thank you so much your interest.
Your comments will increase the quality of our work.
Thank you for your time.
Reviewer 2 Report
I think this is a good study but you have to reorganize the speech better. First correct typos throughout the text, such as: meth-ods, con-text, avail-able, etc.
INTRODUCTION
The introduction is well done but on lines 39 – 43 you repeated several times “it can be said” rephrase the sentences.
MATERIALS AND METHODS
Participants
· Add eligibility and exclusion criteria;
· Delete what is written in lines 116-123 and add it to the Results with a subheading such as (for example) "socio-demographic and anamnestic characteristics."
Measures
· Improve the presentation of measures, before listing the different scales.
RESULTS
Results are well done.
DISCUSSION AND CONCLUSION
Limitations and Future Research
-- In the "limitations and future research" paragraph you have highlighted only the limitations, without saying what future research on the topic might be.
-- Remove the bulleted list to say why the research is important, scale down what you have written, and add it to the end of the "Discussion and Conclusion.".
Author Response
Dear Rewiever,
Firstly, thank you so much your interest and kind.
We have completed the corrections you mentioned on the article.
We are grateful for your valuable comments.
We hope that this article will attract the reader's attention better.
Corrections made to the article:
- The introductory part of the article has been edited as you stated (The introduction is well done but on lines 39 – 43 you repeated several times “it can be said” rephrase the sentences)
- The participants section has been reorganized.
- The "Limitations and Future Research" section has been updated and expanded in line with your suggestions.
If there is anything you would like us to add or correct, we look forward to it.
Best Regards.

Reviewer 3 Report
Thank you for the opportunity to read this very interesting paper. The paper has many strengths that should be of interest to the journal audience. Thus, the following suggestions are around enhancing the presentation for publication and clarifying aspects of the data and reporting.
I will go by line number for the most part. If not, I will try to be as specific as possible in noting the area I am speaking about.
Title
The title is so long. I think that the Title could be shortened and more specific.
Abstract
The abstract should be a single paragraph and should follow the style of structured abstracts, but without headings.
[Page 1, Line 11] If you speak about Background it is not necessary to include the sentence “In this study…”
[Page 1, Line 13] Authors must finish the Background section with the main aim. The aim of this study was to…
[Page 1, Line 14] Authors must specify the type of study design. A cross-sectional study was carried out with a sample… In addition, it is necessary to include the country.
[Page 1, Lines 19-21] Results. Please, provide the values of statistical analysis to the sentence “Findings have shown that emotional reactivity and intolerance of uncer-19 tainty pose a risk for psychological maladjustment in children and adolescents receiving orthodon-20 tic treatment, but psychological resilience has a protective function against this risk”. Where are p-values? (p < .001). Authors must specify it.
Conclusions should be related with the main aim
Keywords. Orthodontic treatment; emotional reactivity; intolerance of uncertainty; psychological adjustment; and mediation relationship are not MeSH Terms.
1. Introduction
[Paragraph 1] Authors must incorporate 1-2 references.
Please provide the expansion of the abbreviations: OCD, REMSEA, CFI, RMR, etc.
2. Materials and Methods
How was the sample chosen? Authors must specify it. : It is necessary to describe the population size, and from this data, provide a calculation of the sample size necessary for the results to be meaningful. It is also necessary to specify the inclusion and exclusion criteria for the study sample.
When did the data get recollected?
Do the authors have a study protocol? The study protocol should be described in detail.
Is the Intolerance of Uncertainty Scale adapted to the Turkish population? Authors must justify their response.
3. Results
Demographic data on respondents should be given
Please, provide the meaning of the variables in the Figures. Figures have low quality.
At last, but not least, I recommend you to make available your data in an open repository. I think it will make this scientific process more transparent, and it allows other researchers to replicate your results.
4. Discussion
Authors need to argue their results better. For instance, in the sentence “It can be said that children with low tolerance of uncertainty will develop more negative reactions in emotional, cognitive and behavioral terms and may disrupt the process considering the factors affecting the duration and success of orthodontic treatment. Therefore, intolerance of uncertainty and emotional reactivity, which are prominent risk factors in orthodontic treatments, are considered to be beneficial as variables that may affect the course of the treatment”
Limitations related with the type of methodology used. Limitations regarding representativeness of respondents should be better addressed Authors must specify it. The fact of having a convenience sample should be included in the limitations of the study.
I recommend to you that include a Conclusion section
References
References could be obsoleted, only 1 of 3 references is current/actual
I wish you all the best.
Author Response
Dear Rewiever,
Firstly, thank you so much your interest and kind.
We have completed the corrections you mentioned on the article.
We are grateful for your valuable comments.
We hope that this article will attract the reader's attention better.
Corrections made to the article:
- We agree that the title of the article is long and Here's how we thought of shortening it. "Investigation of Adjustment Problems in Children Receiving Orthodontic Treatment and the Resilience Factor" Is this ok for you or do you have a suggestion?
- The abstract has been rearranged as you stated.
- Added references to the first paragraph.
- Added expansions of abbreviations (in the Emotional Reactivity Scale)
- The study has not include a study protocol. Because it is not randomized controlled trial. Also, The World Health Organization (WHO) considers the registration of
the study protocol as the publication of an internationally recognized set of information on the design, conduct, and management of clinical trials. Writing a detailed study protocol with all the correct and necessary steps is an important step before starting randomized
controlled trials. - More information was given about the scale (Intolerance scale) and reference was added to the adaptation process.
- The Figures have been rearranged.
- The discussion section has been expanded.
- Limitations updated as per your suggestion
- New references added.
Sincerely

Round 2
Reviewer 3 Report
All my recommendations were added.